# Tissue Expression of Growth Differentiation Factor 11 in Patients with Breast Cancer

**DOI:** 10.3390/diagnostics14070701

**Published:** 2024-03-27

**Authors:** Chia-Chi Chen, Thung-Lip Lee, I-Ting Tsai, Chin-Feng Hsuan, Chia-Chang Hsu, Chao-Ping Wang, Yung-Chuan Lu, Chien-Hsun Lee, Fu-Mei Chung, Yau-Jiunn Lee, Ching-Ting Wei

**Affiliations:** 1Department of Pathology, E-Da Hospital, I-Shou University, Kaohsiung 82445, Taiwan; sasabelievemydream@gmail.com (C.-C.C.); dumbvocal@gmail.com (C.-H.L.); 2School of Medicine, College of Medicine, I-Shou University, Kaohsiung 82445, Taiwan; tsai.iting@gmail.com (I.-T.T.); calvin.hsuan@msa.hinet.net (C.-F.H.); 3Department of Physical Therapy, I-Shou University, Kaohsiung 82445, Taiwan; 4Department of Occupational Therapy, I-Shou University, Kaohsiung 82445, Taiwan; 5Division of Cardiology, Department of Internal Medicine, E-Da Hospital, I-Shou University, Kaohsiung 82445, Taiwan; lip1969@hotmail.com (T.-L.L.); ed100232@livemail.tw (C.-P.W.); chungfumei@gmail.com (F.-M.C.); 6School of Medicine for International Students, College of Medicine, I-Shou University, Kaohsiung 82445, Taiwan; 7Department of Emergency, E-Da Hospital, I-Shou University, Kaohsiung 82445, Taiwan; 8Division of Cardiology, Department of Internal Medicine, E-Da Dachang Hospital, I-Shou University, Kaohsiung 80794, Taiwan; 9Division of Gastroenterology and Hepatology, Department of Internal Medicine, E-Da Hospital, I-Shou University, Kaohsiung 82445, Taiwan; aladarhsu1107@gmail.com; 10Health Examination Center, E-Da Dachang Hospital, I-Shou University, Kaohsiung 80794, Taiwan; 11The School of Chinese Medicine for Post Baccalaureate, College of Medicine, I-Shou University, Kaohsiung 82445, Taiwan; 12Division of Endocrinology and Metabolism, Department of Internal Medicine, E-Da Hospital, I-Shou University, Kaohsiung 82445, Taiwan; gregory.yclu@msa.hinet.net; 13Lee’s Endocrinologic Clinic, Pingtung 90000, Taiwan; lee@leesclinic.org; 14Division of General Surgery, Department of Surgery, E-Da Hospital, I-Shou University, Kaohsiung 82445, Taiwan

**Keywords:** breast cancer, growth differentiation factor 11, expression, clinicopathological characteristics

## Abstract

Protein growth differentiation factor 11 (GDF11) plays crucial roles in cellular processes, including differentiation and development; however, its clinical relevance in breast cancer patients is poorly understood. We enrolled 68 breast cancer patients who underwent surgery at our hospital and assessed the expression of GDF11 in tumorous, ductal carcinoma in situ (DCIS), and non-tumorous tissues using immunohistochemical staining, with interpretation based on histochemical scoring (H-score). Our results indicated higher GDF11 expressions in DCIS and normal tissues compared to tumorous tissues. In addition, the GDF11 H-score was lower in the patients with a tumor size ≥ 2 cm, pathologic T3 + T4 stages, AJCC III-IV stages, Ki67 ≥ 14% status, HER2-negative, and specific molecular tumor subtypes. Notably, the patients with triple-negative breast cancer exhibited a loss of GDF11 expression. Spearman correlation analysis revealed associations between GDF11 expression and various clinicopathological characteristics, including tumor size, stage, Ki67, and molecular subtypes. Furthermore, GDF11 expression was positively correlated with mean corpuscular hemoglobin concentration and negatively correlated with neutrophil count, as well as standard deviation and coefficient of variation of red cell distribution width. These findings suggest that a decreased GDF11 expression may play a role in breast cancer pathogenesis.

## 1. Introduction

Breast cancer is the most common cancer in women worldwide, and it has a substantial impact on global cancer deaths. A previous study reported that there were over 2.3 million new cases and 685,000 deaths in 2020, and that this is estimated to reach over 3 million new cases and 1 million deaths by 2040 [1]. Numerous risk factors contribute to the development of breast cancer, including family history, age, gender, hormonal factors, reproductive factors, lifestyle factors, and genetic mutations (e.g., BRCA1 and BRCA2) [2,3,4,5]. In addition, the proliferative activity of tumor cells is a crucial independent factor associated with the prognosis and treatment response [6,7,8]. Hence, the rapid development of various types of treatment targeting aberrant cell growth, such as cell cycle-targeted chemotherapy for metastatic breast cancer, has enabled specific interventions aimed at tumor cell proliferation pathways. The aims of such treatment are to decrease cellular proliferation and enhance cell death, and at the same time reduce chemotherapy-associated toxicity.

Growth differentiation factor 11 (GDF11), also known as bone morphogenetic protein 11 (BMP11), is a member of the transforming growth factor-β (TGF-β) superfamily. It is associated with the activation of both Smad and non-Smad signaling pathways, thereby regulating target nuclear gene expressions [9,10,11]. Since it was first discovered in 1999, GDF11 has been shown to be involved in normal physiological processes including erythropoiesis and embryonic development [12]. Moreover, GDF11 has been implicated in the pathophysiology of tumor growth [13,14,15], organ development [16], aging [17,18], and the nervous system [19]. GDF11 has been shown to exhibit powerful physiological functions; however, controversy exists regarding its role in cancer biology. For example, GDF11 has been demonstrated to exert tumor suppression effects in some studies such as in patients with triple-negative breast cancer [14,20], but to have the opposite effect in others [21,22]. Nevertheless, the relationships between clinicopathological characteristics and GDF11 expression in patients with breast cancer have yet to be clarified.

Furthermore, previous studies have demonstrated that GDF11 acts as a regulator of erythropoiesis, and it has been associated with the development of mild anemia [23,24]. In addition, there is evidence suggesting that GDF11 may possess immunomodulatory properties, potentially impacting processes such as inflammation and immune cell activity [25,26]. Hence, the aim of this study was to investigate the expression pattern of GDF11 in patients with breast cancer using immunohistochemistry (IHC) in tumorous, ductal carcinoma in situ (DCIS), and non-tumorous tissues. We also explored the associations between clinicopathological characteristics and pretreatment hematological profile with GDF11 expression in these patients.

## 2. Materials and Methods

### 2.1. Study Subjects

This study was conducted at E-Da Hospital from January 2019 to July 2023, and included 68 female patients who underwent surgery for newly diagnosed breast cancer. The inclusion criteria were patients (1) diagnosed with malignant or invasive breast tumors indicative of breast cancer; (2) scheduled for partial mastectomy or mastectomy and had not received any form of cancer treatment including chemotherapy, immunotherapy, or radiotherapy prior to surgery; and (3) who provided consent to participate in the study. The exclusion criteria were patients who (1) had undergone mastectomy or any form of cancer treatment including chemotherapy, radiotherapy, immunotherapy, or other modalities; and (2) refused to provide consent for participation. This study (no. EMRP-111-118) received approval from the Institutional Review Board of E-Da Hospital, and written informed consent was obtained from all of the patients. Information on the patients was extracted from the medical records of the hospital. The mean age of the enrolled patients was 53 (range, 24–86) years. Staging was based on the American Joint Committee on Cancer (AJCC). We chose to define obesity according to the Department of Health, Taiwan, as a BMI of ≥27 kg/m^2^.

### 2.2. Laboratory Measurements

Peripheral blood samples were obtained from the antecubital vein prior to the initiation of any oncological treatment. Peripheral leukocyte analyses were conducted using an automated cell counter (XE-2100 Hematology Alpha Transportation System; Sysmex, Kobe, Japan), and included total leukocyte count, differential neutrophil count, monocyte count, lymphocyte count, and red blood cell (RBC) parameters (hemoglobin, hematocrit, mean corpuscular hemoglobin, mean corpuscular-hemoglobin concentration (MCHC)), platelet count, red cell distribution width-standard deviation (RDW-SD), and RDW-coefficient of variation (RDW-CV). Absolute counts of leukocyte subtypes were calculated as the product of its percentage and total leukocyte count.

### 2.3. Clinicopathologic Characteristics of the Tumors

Histopathological analysis was used to confirm the presence of breast cancer with assessments of estrogen receptor (ER) and progesterone receptor (PR) status. Staging was performed according to the TNM system, and the histological grade was determined using the Bloom–Richardson system. We classified the patients into different groups by tumor size (<2 cm or ≥2 cm), age (<50 years or ≥50 years), lymph node metastasis (N0 + N1 or N2 + N3), pathologic T stage (T0 + T1 or T2 + T3 + T4), histologic grade (1 + 2 or >3), AJCC stage (0–II or III–IV), Ki67 status (<14% or ≥14%), ER (negative or positive), PR (negative or positive), and human epidermal growth factor receptor (HER2) (negative or positive) status. The molecular tumor subtype was determined through IHC analysis of ER, PR, HER2, and the proliferation marker Ki-67 [27]. The studied subtypes were luminal A, luminal B HER2-negative, luminal B HER2-positive, HER2-enriched, and triple-negative. In addition, we define luminal A and B breast cancer subtypes according to the criteria established in a previous study [27]: Luminal A subtype is characterized by ER and PR positivity (≥20%), HER2 negativity, and low Ki-67 expression (<14%). The Luminal B-like subtype, which is HER2 negative, is defined by ER positivity, HER2negativity, and at least one of the following: high Ki-67 expression (≥14%), or negative or low PR expression (<20%). The Luminal B-like subtype with HER2 positivity is characterized by ER positivity, HER2 overexpression or amplification, any level of Ki-67 expression, and any PR expression.

### 2.4. Tissue Sample Collection

We collected samples from all of the enrolled patients, none of whom had undergone chemotherapy or radiotherapy before surgery. Tissue samples from cancerous, DCIS, and adjacent noncancerous areas were obtained, fixed in 10% buffered formalin and embedded in paraffin. For immunohistochemical analysis, 4 μm-thick sections were prepared and stained with hematoxylin and eosin. PR and ER status were examined through IHC staining, and HER2/neu oncoprotein staining was performed using the standard HercepTest procedure (Dako 5204).

### 2.5. GDF11 Expression Analysis

An automated Bond-Max system (Leica Microsystems, Wetzlar, Germany) was used for GDF-11 immunostaining. In brief, the paraffin-embedded formalin-fixed tissue samples were cut into 4-μm-thick sections, and then de-paraffinization was performed with Bond Dewax Solution at 72 °C. Heat-induced epitope retrieval was conducted using the ready-to-use BOND-PRIME Epitope Retrieval Solution 2 (Leica Biosystems, Deer Park, TX, USA) under conditions of pH 9.0 at 100 °C for 20 min, according to the instructions of the manufacturer. Peroxide block was then added to the slides at room temperature for 5 min, and the samples were subsequently incubated with primary rabbit polyclonal antibody against GDF11 (Product # PA5-67058, Invitrogen, Thermo Fisher Scientific, Waltham, MA, USA) at a dilution of 1:100 for 30 min at room temperature. The BOND Polymer Refine Detection system (Product # DS9800) was used with an incubation time of 8 min at room temperature. Color was developed using 3,3′-diaminobenzidine tetrahydrochloride as the chromogen at room temperature for 10 min. Finally, the sections were counterstained with hematoxylin for 5 min, after which the slides were mounted and examined. In addition, a liver cancer sample known to exhibit strong GDF11 expression [28] was utilized as a positive control. For the negative control, the primary antibody was substituted with primary antibody diluent (Tris, Green) (Figure 1). 

### 2.6. Evaluation of Immunohistochemical Staining

GDF11 expression was assessed using both IHC and histochemical scoring (H-score) methods, as described in previous studies [29,30]. Both cytoplasmic and membranous expressions were included and considered positive. The intensity of immunostaining was graded as follows: ‘0’ none, ‘1+’ weakly detectable, ‘2+’ moderate, and ‘3+’ strong (Figure 1). In brief, the H-score was calculated as the sum of the products of the GDF11-positive cell percentage in different staining intensity categories and their respective intensity scores, using the following formula: H-score = [1 × (% of GDF11-positive cells with intensity score 1)] + [2 × (% of GDF11-positive cells with intensity score 2)] + [3 × (% of GDF11-positive cells with intensity score 3)]. The GDF11-positive cell percentage for each intensity category used in the above formula was calculated as the average of 10 randomly chosen high-power fields at 400× magnification. The final H-score ranged from 0 to 300.

### 2.7. Statistical Analysis

Descriptive statistics were used to analyze the data. Data normality was analyzed using the Kolmogorov–Smirnov test, and homogeneity of variance was analyzed using Levene’s test. Continuous variables are expressed as mean ± SD, and categorical variables are expressed as a percentage. The statistical analyses were conducted using JMP version 10.0 for Windows (SAS Institute, Cary, NC, USA). Between-group differences for continuous data were examined using either the Student’s *t*-test or Wilcoxon rank-sum test, as appropriate. Differences in H-scores of GDF11 among different breast cancer status groups were analyzed using one-way analysis of variance, followed by Tukey’s pairwise comparisons. Since the distributions of H-scores of GDF11, white blood cell count, monocyte count, neutrophil count, RBC, hemoglobin, hematocrit, MCH, MCHC, platelet count, RDW-SD, and RDW-CV were skewed, logarithmically transformed values were used for statistical analysis. Associations between H-scores of GDF11 and clinical and biochemical parameters were assessed using Spearman correlation coefficient analysis with a 2-tailed test of significance. *p* values < 0.05 were considered to indicate statistical significance.

## 3. Results

The clinicopathological characteristics of the patients are shown in Table 1. The prevalence rates of hypertension, diabetes mellitus, and hyperlipidemia were 25.0%, 14.7%, and 4.4%, respectively. None of the patients exhibited signs of immunosuppression. In addition, 66.2% of the patients had a tumor size ≥ 2 cm, 20.6% had pathologic T stages of T3 + T4, and 13.2% had lymph node metastasis of N2 + N3. Furthermore, the molecular tumor subtypes luminal A, luminal B HER2-negative, luminal B HER2-positive, HER2-enriched, and triple-negative accounted for 33.82%, 27.94%, 19.12%, 5.88%, and 13.24% of the patients, respectively.

The IHC results demonstrated higher expressions of GDF11 in DCIS and normal tissue specimens compared to tumorous specimens (195.6 ± 41.0 vs. 183.0 ± 58.2 vs. 164.3 ± 64.9, *p* = 0.012, Figure 2). In addition, the IHC analysis revealed GDF11 expression in the cytoplasm and focal membrane of all examined tissues, including breast cancer, DCIS, and non-tumor tissues. Notably, GDF11 expression was higher in DCIS and normal tissue specimens than in tumorous specimens (Figure 3).

We then explored the expression of GDF11 categorized based on categorical variables. The findings revealed a decrease in the H-score of GDF11 in patients with tumor size ≥ 2 cm, pathologic T3 + T4 stages, AJCC III–IV stages, Ki67 ≥ 14% status, HER2-negative, and the molecular tumor subtypes, including luminal B HER2-negative and triple-negative (all *p*-values < 0.05, Table 2). Based on the discrepancy in H-score among molecular subtypes, dummy variables were employed to categorize molecular subtypes, designating luminal B HER2-negative and triple-negative as “1”, and luminal A, luminal B HER2-positive, and HER2-enriched as “0” for Spearman correlation analysis. Spearman correlation analysis showed significant positive correlations between GDF11 H-score with tumor size < 2 cm, pathologic T0 + T1 + T2 stages, and AJCC 0–II stages. In addition, a positive correlation was observed with MCHC, while negative correlations were identified with Ki67, neutrophil count, RDW-SD, RDW-CV, and the molecular tumor subtypes luminal B HER2-negative and triple-negative (Table 3). Furthermore, because Ki-67 and molecular subtypes may have a confounding effect, we employed multiple linear regression analysis to assess the association between GDF11 H-score and the molecular tumor subtypes luminal B HER2-negative and triple-negative, adjusting for Ki-67. The analysis revealed a persistent significant negative association between GDF11 H-score and the mentioned molecular tumor subtypes (β = −0.491, *p* = 0.0002). Moreover, the results of IHC analysis for the localization of GDF11, ER, PR, Her2/neu, and Ki67 in cancer tissues revealed a loss of GDF11 expression in the patients with triple-negative breast cancer (Figure 4).

## 4. Discussion

In this study, we investigated the association between GDF11 and breast cancer, and identified three key findings regarding GDF11 expression patterns, GDF11 H-score, and correlation analysis. First, we found higher expressions of GDF11 in DCIS and normal tissue specimens compared to tumorous specimens. Second, the GDF11 H-scores were lower in the patients with a tumor size ≥ 2 cm, pathologic T3 + T4 stages, AJCC III–IV stages, Ki67 ≥ 14% status, HER2-negative, and the luminal B HER2-negative and triple-negative molecular tumor subtypes. Third, we identified a significant positive correlation between GDF11 H-score and tumor size < 2 cm, pathologic T0 + T1 + T2 stages, and AJCC 0–II stages. In addition, a positive correlation was observed with MCHC, while negative correlations were identified with Ki67, neutrophil count, RDW-SD, RDW-CV, as well as luminal B HER2-negative and triple-negative subtypes.

### 4.1. Higher GDF11 Expressions in DCIS and Normal Tissue Specimens Compared to Tumorous Specimens

The first key finding of this study is the elevated GDF11 expression in DCIS and normal tissue specimens compared to tumorous specimens (Figure 2). This observation may suggest that GDF11 exerts a tumor-suppressive role, aligning with previous studies [13,14,15,20,31]. Liu et al. [13] demonstrated a significant decrease in GDF11 protein expression in pancreatic cancer tissues. In addition, the overexpression of GDF11 was found to suppress aggressive behaviors in pancreatic cancer cells, possibly attributable to its apoptosis-promoting effect on these cells [13]. Gerardo-Ramirez et al. [14] demonstrated the tumor suppressive properties of exogenous GDF11 in hepatocellular carcinoma cells, and Zhang et al. [15] demonstrated that GDF11 exerts its effects through Smad2/3 signaling, and that it plays a tumor suppressor role in liver cancer. In addition, Zhang et al. [31] showed that GDF11 may exert an anti-liver cancer effect by affecting Smad2/3 and inducing apoptosis through the ROS/JNK pathway. Moreover, an in vitro study found that treatment with GDF11 induced cell—cell adhesion and prevented metastasis [20]. Hence, the higher expression of GDF11 in both DCIS and normal tissue specimens in this study may be attributed to the role of GDF11 in regulating cell growth and preventing the progression of pre-cancerous lesions in breast cancer.

GDF11 belongs to the TGF-β protein family, members of which, along with their receptors, are known to play significant roles in regulating cancer. In normal and early carcinoma cells, the TGF-β signaling pathway has a tumor suppression effect, whereas it has been shown to promote cancer metastasis in advanced tumors [32,33,34]. The Human Protein Atlas database shows that GDF11 is implicated in colorectal, liver, breast, and pancreatic cancers. Patients with colorectal cancer and high tumor expressions of GDF11 often have high rates of lymph node metastasis and poor survival [21]. However, in vitro studies have shown that the histone deacetylase inhibitor trichostatin A suppresses tumor growth through the activation of GDF11 [35]. These discrepant results may be attributed to the dual roles of TGF-β at different stages of cancer. Additionally, Alvarez et al. and Auguściak-Duma et al. [36,37] demonstrated that GDF11 is involved in leiomyoma uteri and breast cancer. Therefore, the increased GDF11 expressions in both DCIS and normal tissue specimens in our study may stem from its dual role and context-dependent function [38,39]. Its effects likely vary based on the cancer stage and type, as well as the cellular microenvironment.

### 4.2. H-Score of GDF11 Decreased in Patients with Tumor Size ≥ 2 cm, Pathologic T3 + T4 Stages, and AJCC III–IV Stages

The second key finding of this study is the lower GDF11 H-score in the patients with tumor sizes ≥ 2 cm, pathologic stages T3 + T4, and AJCC stages III–IV (Table 2), which aligns with the findings of a previous study [40]. Wallner et al. reported an association between poor survival and a lower expression of GDF11 and in patients with breast cancer, based on retrospective analysis of tissues [40]. In addition, immunofluorescence analysis of human adenocarcinoma revealed higher expressions of GDF11 in low-stage tumors (G1) compared to fibroadenomas. Interestingly, despite the higher grading (G3) observed in adenocarcinomas, it was associated with lower GDF11 expressions. Moreover, adenocarcinoma (MCF7) and fibroadenoma (MCF10A) cell lines exhibited lower GDF11 mRNA levels in MCF7 cells compared to MCF10A cells.

### 4.3. H-Score of GDF11 and Its Correlations with Ki67

Our results also revealed a lower GDF11 H-score among the patients with a Ki67 ≥ 14% status, accompanied by a significant negative association between GDF11 H-score and Ki67 (Table 2 and Table 3, and Figure 4). These findings align with previous studies [14,41,42]. GDF11 regulates cell proliferation, apoptosis, and differentiation in both normal physiological contexts and pathological conditions [28,43,44]. Gu et al. [41] reported that GDF11 inhibits the apoptosis and proliferation of esophageal cancer cells, and Gerardo-Ramirez et al. [14] reported a significant decrease in cell migration in hepatocellular carcinoma cells treated with GDF11, accompanied by reduced proliferation as judged by Ki67 staining. In addition, Frohlich et al. [42] demonstrated that treating hepatic cancer cells with GDF11 reduced both cell proliferation and apoptosis rates.

### 4.4. Loss of GDF11 Expression in Patients with Triple-Negative Breast Cancer

We also found a lower GDF11 H-score in the patients with the triple-negative molecular tumor subtype (Table 2 and Figure 4). Bajikar et al. [20] reported that GDF11 had a tumor suppressive effect in triple-negative breast cancer [20]. In addition, they showed that acutely stimulating triple-negative breast cancer cells (MDA-MB-231) and immortalized human mammary epithelial cells (MCF10A-5E) with GDF11 (250 ng/mL) induced the phosphorylation of Smad2/3, while Smad1/5 remained unaffected, suggesting that the Smad2/3 pathway has a central role in the mammary epithelium [20]. They also demonstrated that GDF11 signaling in triple-negative breast cancer cells led to modifications in multicellular organization, resulting in a stabilized epithelial state. Moreover, xenografts with occluded mammary ducts co-injected with 200 ng rGDF11 exhibited reduced bioluminescence of triple-negative breast cancer and decreased cell proliferation without affecting apoptosis compared to xenografts without the co-injection of rGDF11 [20]. Furthermore, the signaling pathways in triple-negative breast cancer cell lines were impaired despite the increased levels of GDF11. This impairment stemmed from a loss of GDF11 function in the triple-negative breast cancer cells, which was associated with a disruption in maturation and secretion noted in seven cell lines. A loss of function of the proprotein convertase subtilisin/kexin type 5 was also noted in the triple-negative breast cancer cells, which is responsible for the maturation of proGDF11 into its active peptide form. This deficiency was accompanied by the extracellular accumulation of immature proGDF11 and loss of GDF11 activity within the cells [20].

### 4.5. Correlations between H-Score of GDF11 and MCHC, Neutrophil Count, RDW-SD, and RDW-CV

The third key finding of this study is the positive correlation between GDF11 H-score with MCHC, and the negative correlations between GDF11 H-score with neutrophil count, RDW-SD, and RDW-CV. Previous research has indicated that the downregulation of GDF11 expression has been proposed as a potential treatment for thalassemia [45]. Han et al. [46] further reported that the overexpression of GDF11 in the bloodstream of patients with myelodysplastic syndrome could hinder the production of RBCs, thereby exacerbating the patients’ condition. However, in the current study, an elevated GDF11 H-score was correlated with a higher MCHC level. This suggests that GDF11 may induce alterations in erythropoiesis or changes in RBC morphology within the context of breast cancer. Further investigations are warranted to clarify discrepancies among studies. Furthermore, neutrophils are a type of white blood cell crucial for immune defense against infections. RDW-SD serves as a measure of variability in the size of RBCs, with elevated levels potentially associated with poor nutritional status [47], inflammation [48], and oxidative stress [49]. Furthermore, alterations in the distribution of RBCs may indicate changes in the tumor microenvironment, a critical factor in supporting the survival and growth of cancer cells [50]. In addition, Li et al. [51] found a correlation between RDW-CV and the pathological features of colorectal cancer, suggesting a more malignant nature of the tumor. In the present study, despite observing negative correlations between GDF11 H-score and neutrophil count, RDW-SD, and RDW-CV in the breast cancer patients, further investigations are warranted to clarify systemic GDF11 levels in these patients, as they were not measured in this study.

### 4.6. Contributions and Limitations

Our study contributes to the literature in three significant ways. First, GDF11 expression patterns: the observation of higher expressions in DCIS and normal tissue specimens compared to tumorous specimens suggests that GDF11 potentially plays a role in the early stages of breast cancer development. This finding could contribute to identifying GDF11 as a potential biomarker for early detection or as a target for intervention strategies aimed at preventing tumor progression. Second, GDF11 H-score: the lower GDF11 H-score in the patients with larger tumor sizes, advanced pathological stages, and specific molecular subtypes of breast cancer (such as luminal B HER2-negative and triple-negative subtypes) implies that GDF11 exerts a potential tumor-suppressive role. Understanding the molecular mechanisms underlying this decrease in GDF11 H-score could lead to the development of targeted therapies or prognostic tools for patients with these aggressive subtypes. Third, correlation analysis: the significant positive correlation between GDF11 H-score and favorable prognostic factors, such as smaller tumor size and earlier pathological stages, hints at a potential protective or regulatory role for GDF11 in breast cancer progression, given the absence of survival analysis regarding GDF11 in this study. Overall, our findings contribute to our understanding of the complex interplay between GDF11 expression and breast cancer progression, potentially paving the way for the development of novel diagnostic, prognostic, and therapeutic approaches in breast cancer management. Further research into the underlying mechanisms and validation in larger cohorts are needed to confirm these contributions and translate them into clinical applications.

While our study provides valuable insights, it is important to acknowledge its limitations. Primarily, the sample size was relatively small. Future investigations should aim to include larger cohorts to validate our findings more robustly. Another limitation of this study is that we exclusively relied on IHC for analysis. Further research employing diverse methodologies is warranted to fully elucidate the role of GDF11 in breast cancer. Furthermore, our study lacks detailed information on the molecular mechanisms underlying the function of GDF11 in breast cancer pathogenesis. To address this gap, future studies leveraging well-known cancer genome databases such as The Cancer Genome Atlas and Gene Expression Omnibus, or in vitro studies using established breast cancer cell lines such as T-47D, SkBr3, MDA-MB-231, and MCF-7 are warranted.

## 5. Conclusions

Our results demonstrated a lower GDF11 H-score among patients with larger tumor sizes, advanced stages, higher Ki67 levels, HER2-negative, and specific molecular tumor subtypes. Furthermore, GDF11 expression was absent in patients with triple-negative breast cancer. These findings suggest a potential role for decreased GDF11 expression in breast cancer pathogenesis. Further investigations are warranted to elucidate the precise mechanisms underlying GDF11 signaling in breast cancer development. Moreover, exploration of GDF11 as a potential therapeutic target and diagnostic marker may lead to the development of novel treatment strategies.

## Figures and Tables

**Figure 1 diagnostics-14-00701-f001:**
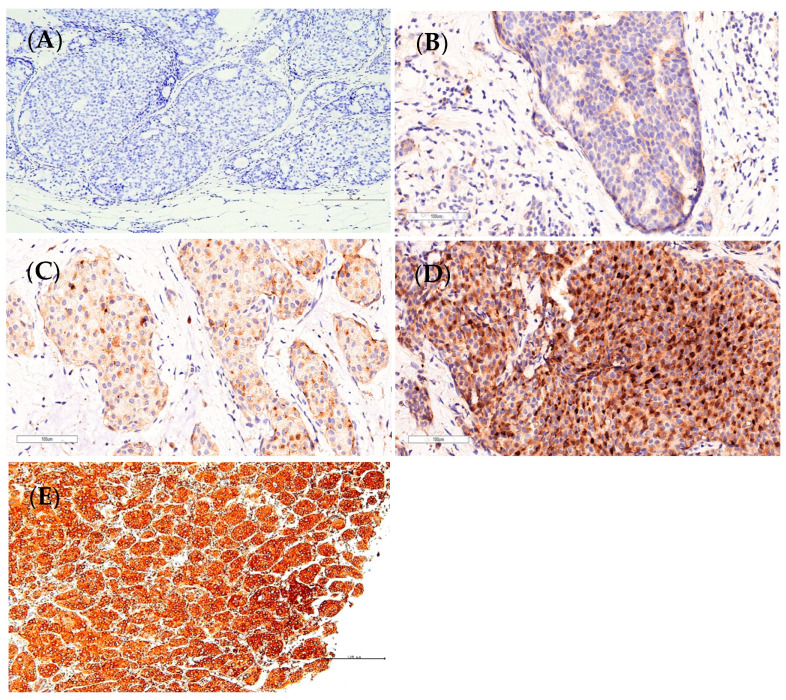
Assessment of growth differentiation factor 11 (GDF11) staining was conducted. The intensity of GDF11 staining within breast cancer specimens was evaluated using immunohistochemistry. Representative figures illustrating negative control (**A**), weak 1+ (**B**), moderate 2+ (**C**), strong staining 3+ (**D**), and positive control (**E**) for liver cancer tissue are provided.

**Figure 2 diagnostics-14-00701-f002:**
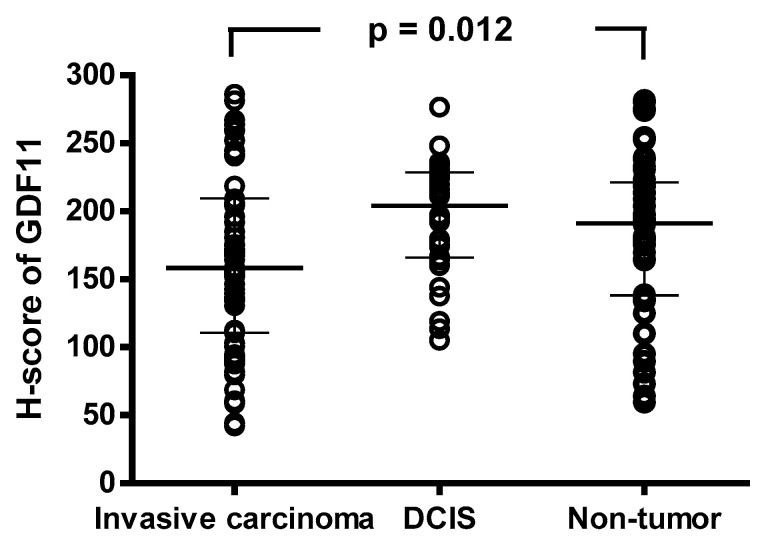
Associations between H-score of GDF11 and breast cancer status were investigated. Bars represent the median (interquartile range), and differences between the groups were analyzed using one-way analysis of variance. GDF11, growth differentiation factor 11; DCIS, ductal carcinoma in situ; H-score, histochemical scoring.

**Figure 3 diagnostics-14-00701-f003:**
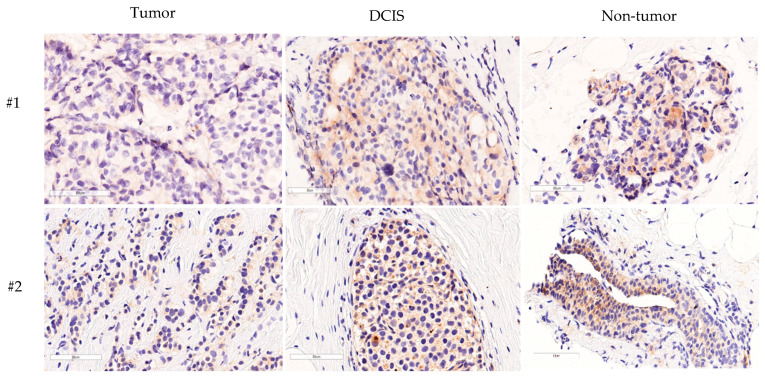
The immunohistochemistry analysis revealed the expression of growth differentiation factor 11 (GDF11) in the cytoplasm and focal membrane of all examined tissues, including breast cancer, ductal carcinoma in situ (DCIS), and non-tumor tissues. Interestingly, the GDF11 expression was higher in the DCIS and normal tissue specimens compared to the tumorous specimens (#1 and #2).

**Figure 4 diagnostics-14-00701-f004:**
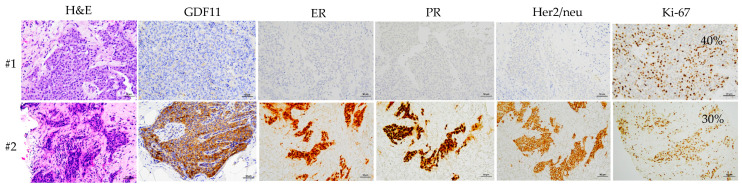
Illustrative immunohistochemistry images are provided to depict the localization of H&E, GDF11, ER, PR, Her2/neu, and Ki67 in two cancer tissues (#1 and #2). H&E, hematoxylin and eosin stain; GDF11, growth differentiation factor 11; ER, estrogen receptor; PR, progesterone receptor.

**Table 1 diagnostics-14-00701-t001:** The clinicopathological characteristics of the 68 patients diagnosed with breast cancer.

Parameter	Number	Percent
Age (years)		
<50	32	47.1
≥50	36	52.9
Range	24–86	
Mean ± SD	52.9 ± 13.5	
Obesity	25	36.8
Menstrual status		
Pre-menopause	32	47.1
Post-menopause	36	52.9
Comorbidities		
Hypertension	17	25.0
Diabetes mellitus	10	14.7
Hyperlipidemia	3	4.4
Tumor size (cm)		
<2	23	33.8
≥2	45	66.2
Pathologic T stage		
T0 + T1 + T2	54	79.4
T3 + T4	14	20.6
Lymph node metastasis		
N0 + N1	59	86.8
N2 + N3	9	13.2
Histologic grade		
1 + 2	39	57.4
>3	29	42.6
AJCC Stage		
0–II	47	69.1
III–IV	21	30.9
Ki67 status		
<14%	26	38.2
≥14%	42	61.8
Estrogen receptor		
Negative	14	20.6
Positive	54	79.4
Progesterone receptor		
Negative	24	35.3
Positive	44	64.7
HER2		
Negative	51	75.0
Positive	17	25.0
Molecular tumor subtype		
1 (Luminal A)	23	33.82
2 (Luminal B HER2-negative)	19	27.94
3 (Luminal B HER2-positive)	13	19.12
4 (HER2-enriched)	4	5.88
5 (Triple-negative)	9	13.24

AJCC, American Joint Committee on Cancer.

**Table 2 diagnostics-14-00701-t002:** Expression of growth differentiation factor 11 grouped according to categorical variables.

Parameter	*N*	H-Score	*p*-Value
Age (years)			
<50	32	159.6 ± 65.7	0.581
≥50	36	168.4 ± 64.8	
Tumor size (cm)			
<2	23	204.9 ± 58.3	0.0001
≥2	45	143.5 ± 58.4	
Pathologic T stage			
T0 + T1 + T2	54	172.1 ± 63.9	0.049
T3 + T4	14	134.1 ± 61.7	
Lymph node metastasis			
N0 + N1	59	165.6 ± 62.3	0.550
N2 + N3	9	155.6 ± 84.1	
Histologic grade			
1 + 2	39	174.8 ± 63.8	0.089
>3	29	150.1 ± 64.7	
AJCC stage			
0–II	47	179.8 ± 65.8	0.030
III–IV	21	138.6 ± 60.6	
Ki67 status			
<14%	26	193.7 ± 61.9	0.004
≥14%	42	146.0 ± 62.0	
Estrogen receptor			
Negative	14	162.9 ± 63.3	0.923
Positive	54	164.8 ± 66.4	
Progesterone receptor			
Negative	24	154.2 ± 62.4	0.358
Positive	44	169.8 ± 67.0	
HER2			
Negative	51	155.8 ± 64.8	0.048
Positive	17	189.8 ± 60.0	
Molecular tumor subtype			
Luminal A	23	188.5 ± 60.1	0.002
Luminal B HER2-negative	19	122.4 ± 53.7	
Luminal B HER2-positive	13	184.0 ± 65.2	
HER2-enriched	4	208.8 ± 39.6	
Triple-negative	9	142.5 ± 62.5	

Data are mean ± SD. AJCC, American Joint Committee on Cancer.

**Table 3 diagnostics-14-00701-t003:** Spearman correlation analysis of clinical and biochemical variables with expression of growth differentiation factor 11.

	H-Score of GDF11	
Parameter	r	*p*-Value
Tumor size (<2 cm versus ≥2 cm)	0.436	0.0002
Pathologic T stage (T0 + T1 + T2 versus T3 + T4)	0.239	0.049
AJCC stage (0–II versus III–IV)	0.258	0.048
Ki67	−0.256	0.038
Luminal B HER2-negative and triple-negative versus luminal A, luminal B HER2-positive, and HER2-enriched	−0.370	0.002
White blood cell count	−0.157	0.205
Neutrophil count	−0.291	0.037
Monocyte count	−0.168	0.234
Lymphocyte count	0.157	0.267
Red blood cells	0.032	0.796
Hemoglobin	0.159	0.199
Hematocrit	0.074	0.554
MCH	0.212	0.086
MCHC	0.418	0.0004
Platelet count	−0.096	0.439
RDW-SD	−0.269	0.028
RDW-CV	−0.248	0.043

GDF11, growth differentiation factor 11; AJCC, American Joint Committee on Cancer; MCH, mean corpuscular hemoglobin; MCHC, mean corpuscular-hemoglobin concentration; RDW, red cell distribution width; SD, standard deviation; CV, coefficient of variation. Molecular tumor subtype: (Luminal A), (Luminal B HER2-negative), (Luminal B HER2-positive), (HER2-enriched), and (Triple-negative).

## Data Availability

The datasets used and/or analyzed during the current study are available from the corresponding author upon reasonable request.

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
