# Peer review of "Tissue Expression of Growth Differentiation Factor 11 in Patients with Breast Cancer"

_diagnostics, 2024, doi:10.3390/diagnostics14070701_

Round 1

Reviewer 1 Report

Comments and Suggestions for Authors

In this study researchers detected the level of GDF11 in DISC and in normal and tumorous tissues. Although the GDF11 expression was earlier written and published in breast cancer and other cancer types as well (Simoni-Nieves A, Gerardo-Ramírez M, Pedraza-Vázquez G, Chávez-Rodríguez L, Bucio L, Souza V, Miranda-Labra RU, Gomez-Quiroz LE, Gutiérrez-Ruiz MC. GDF11 Implications in Cancer Biology and Metabolism. Facts and Controversies. Front Oncol. 2019 Oct 15;9:1039,  Bajikar SS, Wang CC, Borten MA, Pereira EJ, Atkins KA, Janes KA. Tumor-Suppressor Inactivation of GDF11 Occurs by Precursor Sequestration in Triple-Negative Breast Cancer. Dev Cell. 2017 Nov 20;43(4):418-435.e13.,  Liu Y, Shao L, Chen K, Wang Z, Wang J, Jing W, Hu M. GDF11 restrains tumor growth by promoting apoptosis in pancreatic cancer. Onco Targets Ther. 2018;11:8371-8379). Certainly the results came from their papients samples. The experimental conditions are well written, the results are nicely presented.

Author Response

  1. In this study researchers detected the level of GDF11 in DISC and in normal and tumorous tissues. Although the GDF11 expression was earlier written and published in breast cancer and other cancer types as well (Simoni-Nieves A, Gerardo-Ramírez M, Pedraza-Vázquez G, Chávez-Rodríguez L, Bucio L, Souza V, Miranda-Labra RU, Gomez-Quiroz LE, Gutiérrez-Ruiz MC. GDF11 Implications in Cancer Biology and Metabolism. Facts and Controversies. Front Oncol. 2019 Oct 15;9:1039, Bajikar SS, Wang CC, Borten MA, Pereira EJ, Atkins KA, Janes KA. Tumor-Suppressor Inactivation of GDF11 Occurs by Precursor Sequestration in Triple-Negative Breast Cancer. Dev Cell. 2017 Nov 20;43(4):418-435.e13., Liu Y, Shao L, Chen K, Wang Z, Wang J, Jing W, Hu M. GDF11 restrains tumor growth by promoting apoptosis in pancreatic cancer. Onco Targets Ther. 2018;11:8371-8379). Certainly the results came from their papients samples. The experimental conditions are well written, the results are nicely presented.

Author Response: Thank you for reviewer comment.

Reviewer 2 Report

Comments and Suggestions for Authors

Deciphering the clinical role of potential biomarkers in cancers is always interesting. However, I notice many inconsistencies and even incompleteness.

Starting with the conservation of biological material, at first, the authors mention preservation by temperature (line 92), but, later, they mention the FFPE procedure (lines 117-119). This is somewhat contradictory.

Regarding the analysis of peripheral blood, the authors do not mention whether the collection was carried out before or after any oncological treatment, unlike the description they make for the collection of tumor tissue.

Nor do they mention whether patients with any type of underlying disease, which could imply important differences, such as immunosuppression, etc., were excluded. The authors make no mention of inclusion or exclusion criteria.

Regarding statistical analyses, there are several inconsistent points. Although they mention that "All variables underwent examination for descriptive data", I have doubts about the distributions of continuous data, especially from GDF11. Adherence to the Normal distribution is an indispensable prerequisite for applying tests known as "parametric" (Student's t-test and ANOVA), and carrying out such tests on "non-parametric" data implies a type I error. authors do not mention whether they checked other prerequisites, such as homogeneity of variances, etc.

Regarding statistics, the authors applied tests in situations that break their assumptions. For example, correlations must be applied between continuous or at least ordinal variables. In the case of dichotomous categorical variables, analyses of differences in means/medians or distributions must be performed. Even if there is a case in which an order is perceived, such as Ki-67 <14% and ≥14%, it does not make sense to apply a correlation in the categorization instead of using the original continuous variable. However, this is not the case for a correlation with molecular subtypes, where there is no order.

It is possible to see that the authors carried out many analyses with highly redundant variables. For example, classifying tumors as <2cm and ≥2cm or T0+T1 and T2+T3+T4 is very redundant. Most T4 tumors are ≥2 cm, and all T2 and T3 are ≥2 cm; just see the frequencies in Table 1 for such variables.

Regarding classifications, the authors provide a reference for the classification into molecular subtypes (surrogate) (DOI:10.1016/j.semcancer.2020.03.014). However, this reference does not provide any information about ER, PR, or Ki-67 cutoff points for this purpose. Authors must use currently accepted consensus statements (DOI:10.1093/annonc/mdt303; 10.1093/annonc/mdv221).

In the Discussion, the authors make strong assumptions about the role of GDF11 in tumor progression and systemic effects (lines 352-355 and 371-374). Correlation and causation are very different matters. The authors did not perform any experiments, such as silencing or inhibiting GDF11, nor even survival analyzes (which are still very correlational), to confirm that GDF11 has a role in tumor progression. The authors also did not measure systemic GDF11. It is known that what is observed locally in the tumor may not be related to the systemic level.

Still regarding correlations, what was observed for Ki-67 cutoffs and molecular subtypes may have a confounding effect. Luminal B, HER2-enriched, and triple-negative subtypes are mostly Ki-67-high. To be able to improve conclusions, multivariate analyses would be necessary.

Still in the Discussion, the authors present methodology data from other studies with a completely different experimental design that is not relevant, as in lines 293 to 295.

As minor points, I point out that: the Introduction makes no mention of the role of GDF11 with the figurative elements of blood or immunological function, leaving only to bring some information in the Discussion; the authors do not present a table with clinicopathological data of the patients.

Comments on the Quality of English Language

Many sentences are poorly written, for example:

"Additionally, adenocarcinoma (MCF7) and fibroadenoma (MCF10A) cell lines exhibited lower levels of GDF11 mRNA in MCF7 cells compared to MCF10A cells."

Check table titles, such as Table 1:

“Illustrate the clinicopathological characteristics of 68 patients with breast cancer”

Author Response

Response to Reviewer 2:

Deciphering the clinical role of potential biomarkers in cancers is always

interesting. However, I notice many inconsistencies and even incompleteness.

  1. Starting with the conservation of biological material, at first, the authors mention preservation by temperature (line 92), but, later, they mention the FFPE procedure (lines 117-119). This is somewhat contradictory.

  Author Response: Thank you for your reviewer comment. We acknowledge that the sentences "Tumorous and non-tumorous tissue specimens of the resected breast were preserved at -80˚C for subsequent assays (line 92)" contained an error. These sentences have been removed from the revised manuscript. For further reference, please consult page 3, lines 102-103 of the revised manuscript.

  1. Regarding the analysis of peripheral blood, the authors do not mention whether the collection was carried out before or after any oncological treatment, unlike the description they make for the collection of tumor tissue.

  Author Response: In the present study, peripheral blood samples were obtained from the antecubital vein prior to the initiation of any oncological treatment. Additional sentences have been incorporated into the Materials and Methods section. For specific details, please refer to page 3, lines 105-106 of the revised manuscript.

  1. Nor do they mention whether patients with any type of underlying disease, which could imply important differences, such as immunosuppression, etc., were excluded. The authors make no mention of inclusion or exclusion criteria.

  Author Response: In the present study, the prevalence rates of comorbidities such as hypertension, diabetes mellitus, and hyperlipidemia among the patients were 25.0%, 14.7%, and 4.4%, respectively. Additionally, none of the patients exhibited signs of immunosuppression. We have revised Table 1 and added sentences in the Results section. Please refer to Table 1 on page 5 and page 5, lines 195-197 of the revised manuscript for the updated data. Furthermore, inclusion and exclusion criteria have been incorporated into the Materials and Methods section. Please see page 2, lines 90-96 of the revised manuscript for details.

  1. Regarding statistical analyses, there are several inconsistent points. Although they mention that "All variables underwent examination for descriptive data", I have doubts about the distributions of continuous data, especially from GDF11. Adherence to the Normal distribution is an indispensable prerequisite for applying tests known as "parametric" (Student's t-test and ANOVA), and carrying out such tests on "non-parametric" data implies a type I error. authors do not mention whether they checked other prerequisites, such as homogeneity of variances, etc.

  Author Response: In the present study, all variables underwent examination for descriptive data. Data normality was analyzed using the Kolmogorov- Smirnov test. Homogeneity of variances was analyzed using Levene's Test. Furthermore, since the distributions of H-scores of GDF11, white blood cell count, monocyte count, neutrophil count, RBC, hemoglobin, hematocrit, MCH, MCHC, platelet count, RDW-SD, and RDW-CV were skewed, logarithmically transformed values were used for statistical analysis. We have included additional sentences in the Statistical Analysis section to address this point. Please refer to page 5, lines 179-181, and lines 187-190 of the revised manuscript for further details.

  1. Regarding statistics, the authors applied tests in situations that break their assumptions. For example, correlations must be applied between continuous or at least ordinal variables. In the case of dichotomous categorical variables, analyses of differences in means/medians or distributions must be performed. Even if there is a case in which an order is perceived, such as Ki-67 <14% and ≥14%, it does not make sense to apply a correlation in the categorization instead of using the original continuous variable. However, this is not the case for a correlation with molecular subtypes, where there is no order.

  Author Response: Based on reviewer comments, we reverted the categorization of Ki-67 status (<14% versus ≥14%) back to the original continuous variable. Additionally, based on the discrepancy in H-score among molecular subtypes, dummy variables were employed to categorize molecular subtypes, designating luminal B HER2-negative and triple-negative as “1”, and luminal A, luminal B HER2-positive, and HER2-enriched as “0” for Spearman correlation analysis. The findings revealed a significant negative correlation between GDF11 H-score and the molecular tumor subtypes luminal B HER2-negative and triple-negative. The Abstract and Results sections have been revised accordingly. For further details, please refer to Table 3 on page 9, page 1, line 41, page 7, lines 236-239, lines 241-244, and page 8, line 1 of the revised manuscript.

  1. It is possible to see that the authors carried out many analyses with highly redundant variables. For example, classifying tumors as <2cm and ≥2cm or T0+T1 and T2+T3+T4 is very redundant. Most T4 tumors are ≥2 cm, and all T2 and T3 are ≥2 cm; just see the frequencies in Table 1 for such variables.

  Author Response: Based on reviewer comments, we have revised the classification of pathologic T stage from “T0+T1 and T2+T3+T4” to “T0+T1+T2 and T3+T4” in Tables 1-3. The Abstract and Results sections have been updated accordingly. For further details, please refer to Tables 1-3 on pages 6, 8, and 9, as well as page 1, line 38, page 5, lines 198-199, page 7, lines 234 and 241 of the revised manuscript.

  1. Regarding classifications, the authors provide a reference for the classification into molecular subtypes (surrogate) (DOI:10.1016/j.semcancer.2020.03.014). However, this reference does not provide any information about ER, PR, or Ki-67 cutoff points for this purpose. Authors must use currently accepted consensus statements (DOI:10.1093/annonc/mdt303; 10.1093/ annonc/mdv221).

  Author Response: Thank you for the reminder. In the present study, the classification of the molecular tumor subtype was based on the paper by Goldhirsch et al. (DOI: 10.1093/annonc/mdt303). Unfortunately, we cited the wrong paper in original reference 23. We have revised reference 27 (Rakha, E.A.; Pareja, F.G. New Advances in Molecular Breast Cancer Pathology. Semin. Cancer Biol. 2021, 72, 102-113.) to (Goldhirsch, A.; Winer, E.P.; Coates, A.S.; Gelber, R.D.; Piccart-Gebhart, M.; Thürlimann, B.; Senn, H.J.; Panel members. Personalizing the treatment of women with early breast cancer: highlights of the St Gallen International Expert Consensus on the Primary Therapy of Early Breast Cancer 2013. Ann. Oncol. 2013, 24, 2206-2223) of the revised manuscript.

  1. In the Discussion, the authors make strong assumptions about the role of GDF11 in tumor progression and systemic effects (lines 352-355 and 371-374). Correlation and causation are very different matters. The authors did not perform any experiments, such as silencing or inhibiting GDF11, nor even survival analyzes (which are still very correlational), to confirm that GDF11 has a role in tumor progression. The authors also did not measure systemic GDF11. It is known that what is observed locally in the tumor may not be related to the systemic level.

  Author Response: Thank you for the reviewer's comment. Indeed, in the present study, we did not perform any experiments such as silencing or inhibiting GDF11, measure systemic GDF11, nor conduct survival analyses to confirm the role of GDF11 in systemic effects and tumor progression. Consequently, we have revised our discussion concerning the role of GDF11 in tumor progression and systemic effects in Discussion section. For further details, please refer to page 12, lines 396-402, and page 13, lines 417-423 of the revised manuscript.

  1. Still regarding correlations, what was observed for Ki-67 cutoffs and molecular subtypes may have a confounding effect. Luminal B, HER2-enriched, and triple-negative subtypes are mostly Ki-67-high. To be able to improve conclusions, multivariate analyses would be necessary.

  Author Response: We employed multiple linear regression analysis to assess the association between GDF11 H-score and the molecular tumor subtypes luminal B HER2-negative and triple-negative, adjusting for Ki-67. The analysis revealed a persistent significant negative association between GDF11 H-score and the mentioned molecular tumor subtypes (β = -0.491, p = 0.0002). We added sentences to address this result in Results section. Please see page 8, lines 245-251 of the revised manuscript for further details.

  1. Still in the Discussion, the authors present methodology data from other studies with a completely different experimental design that is not relevant, as in lines 293 to 295.

  Author Response: We have excluded the sentences, which stated, "Additionally, the administration of GDF11 (2 μg/mL) to MCF7 cells for 7 days resulted in a modest decrease in migratory capacity, as observed in the scratch assay," from the revised manuscript in accordance with the reviewer's suggestion. Please see page 11, lines 339-340 of the revised manuscript for further details.

  1. As minor points, I point out that: the Introduction makes no mention of the role of GDF11 with the figurative elements of blood or immunological function, leaving only to bring some information in the Discussion; the authors do not present a table with clinicopathological data of the patients.

  Author Response: We have incorporated additional sentences in the Introduction section to discuss the role of GDF11 in relation to blood or immunological functions. Please see page 2, lines 76-80 of the revised manuscript for further details. Furthermore, clinicopathological data has been included in Table 1. Moreover, we have supplemented patient clinical data, including information on obesity, menstrual status, and comorbidities, in Table 1 of the revised manuscript. Please see Table 1 on page 5 of the revised manuscript.

  1. Comments on the Quality of English Language

   Many sentences are poorly written, for example:

  "Additionally, adenocarcinoma (MCF7) and fibroadenoma (MCF10A) cell lines exhibited lower levels of GDF11 mRNA in MCF7 cells compared to MCF10A cells."

  Author Response: We have revised "Additionally, adenocarcinoma (MCF7) and fibroadenoma (MCF10A) cell lines exhibited lower levels of GDF11 mRNA in MCF7 cells compared to MCF10A cells." to “Moreover, adenocarcinoma (MCF7) and fibroadenoma (MCF10A) cell lines exhibited lower GDF11 mRNA levels in MCF7 cells compared to MCF10A cells.” Please see page 11, lines 337-338 of the revised manuscript. In addition, we have thoroughly revised the manuscript, and an English lecturer has conducted a review to ensure its accuracy and clarity. Please find the Medical Editing Certificate below.

  1. Check table titles, such as Table 1:

“Illustrate the clinicopathological characteristics of 68 patients with breast cancer”

  Author Response: We have revised the title of Table 1 to read as follows: “The clinicopathological characteristics of the 68 patients diagnosed with breast cancer”. Please see page 5, lines 204-205 of the revised manuscript. Additionally, all table titles have been reviewed for accuracy.

Reviewer 3 Report

Comments and Suggestions for Authors

Tissue expression of growth differentiation factor 11 in patients with breast cancer

Chia-chi Chen and co-author present an extensively reported immunohistochemical analysis of GDF11 on 68 breast cancer patients correlated with clinicopathological parameters. They essentially found a higher expression in benign tissue and tumoral tissue with a more favourable biology.

1)     GDF11 immunohistochemistry (p/2/15 85-92; p3/15: 124-138; Fig2 p6/15).

a)      The clone stated in the study is for immunohistochemical use, but not necessary for FFPE-tissue as you state on p3/15, section 2.5. This might be one factor for the faint staining. An alternative from the same company is PA5-11928 which is for use in FFPE.

b)     Please, specify that you used a rabbit polyclonal antibody

c)      Moreover, it is not clear, whether you finally used FFPE or frozen tissue since you stated on p 2/15 that tumour tissue preserved at – 80°C was used for subsequent analysis.

d)     In the staining protocol rather use the temperature (eg 36°C) instead of room temperature.

e)     Due to the very faint staining the use of a proper positive control (eg. Testis with positive Leydig cells) and also negative controls should be shown.

f)       The immunostainings in Figure 2 are not convincing. They especially do not allow distinguishing a back ground staining from proper positivity. Since we normally show the best pictures available, the whole study is doubtful, based on these staining results.

g)      Please, provide proper staining results with positive, negative control, faint, moderate and strong immunostainings additionally to what you have already shown.

2)     Study cohort (p3/15 2.3.clinicopathological tumour characteristics ; p4/15: 3. Results 164-169)

a)      The study cohort is low with 68 patients.

b)     The percentage of HER2-positive patients is very high with 40%. The range of HER-2 positive cases in a cohort of newly diagnosed breast cancer patients is around 12-15%. Can you state on that?

c)      How did you define luminal A and B (Ki67, ER/PR)?

3)     Correlation with blood parameters

a)      Correlation with blood parameters should be omitted in this study, since it does not bring additional information to the key findings and the conclusions on the association with GDF11 are very hypothetic.

4)     Manuscript text

The text is too long and there are too many redundancies. Please, focus on the key issues and results.

Author Response

Response to Reviewer 3:

  • GDF11 immunohistochemistry (p/2/15 85-92; p3/15: 124-138; Fig2 p6/15).

(a) The clone stated in the study is for immunohistochemical use, but not necessary for FFPE-tissue as you state on p3/15, section 2.5. This might be one factor for the faint staining. An alternative from the same company is PA5-11928 which is for use in FFPE.

Author Response: Thank you for your kind advice. It is true that PA5-11928 is a better choice for use in GDF 11 FFPE tissue immunohistochemistry. In our experiment, we followed the standard FFPE immunohistochemistry staining protocol suggested by Thermo Fisher Scientific. Meanwhile, we also repeated each sample thrice to ensure reproducibility. Additionally, we searched for GDF 11 antibodies on Thermo Fisher Scientific's website and found that PA5-67058 was the only antibody used in immunohistochemistry with further advanced verification (https://www.thermofisher.com/antibody/primary/ target/gdf11). As a result, although PA5-67058 might not be the best antibody for use in FFPE tissue, it still could effectively represent GDF 11 expression in our experiments with minimal influence.

(b) Please, specify that you used a rabbit polyclonal antibody

Author Response: We included information on the utilization of the rabbit polyclonal antibody in the Materials and Methods section. For further details, please refer to page 4, line 150 of the revised manuscript.

(c) Moreover, it is not clear, whether you finally used FFPE or frozen tissue since you stated on p 2/15 that tumour tissue preserved at -80°C was used for subsequent analysis.

Author Response: Thank you for your reviewer comment. We acknowledge that the sentences "Tumorous and non-tumorous tissue specimens of the resected breast were preserved at -80˚C for subsequent assays (line 92)" contained an error. These sentences have been removed from the revised manuscript. For further reference, please consult page 3, lines 102-103 of the revised manuscript.

(d) In the staining protocol rather use the temperature (eg 36°C) instead of room temperature.

Author Response: The datasheet for the GDF11 Rabbit Polyclonal Antibody (PA5-67058), accessible at thermofisher.com, does not stipulate the necessity of utilizing a temperature of 36°C in the staining protocol. Initially, we attempted a temperature of 37°C in the staining protocol, but observed a very weak GDF11 expression. Subsequently, upon utilizing room temperature, there was a notable increase in the staining expression of GDF11. Consequently, for the current study, we have opted to employ room temperature in the staining protocol for the analysis of GDF11 expression.

(e) Due to the very faint staining the use of a proper positive control (eg. Testis with positive Leydig cells) and also negative controls should be shown.

Author Response: A liver cancer sample known to exhibit strong GDF11 expression (1) was utilized as a positive control. For the negative control, the primary antibody was substituted with primary antibody diluent (Tris, Green). Explanatory sentences were added in the Materials and Methods section to detail the methodologies employed for positive and negative control. Please see Figure 1 on page 4 and lines 157-159 of the revised manuscript.

(f) The immunostainings in Figure 2 are not convincing. They especially do not allow distinguishing a back ground staining from proper positivity. Since we normally show the best pictures available, the whole study is doubtful, based on these staining results.

Author Response: It is acknowledged that discerning changes in immunohistochemistry staining can pose challenges to visual detection. While disparities in GDF11 expression between non-tumor and tumor tissues are observable under microscopic examination, we employ histochemical scoring (H-score) methodologies to meticulously assess result accuracy, as corroborated by prior scholarly investigations (2, 3). First, we define both cytoplasmic and membranous expressions were included and considered positive. Then, the intensity of immunostaining was graded as follows: '0' none, '1+' weakly detectable, '2+' moderate, and '3+' strong. Finally, the H-score was calculated as the sum of the products of the GDF-11-positive cell percentage in different staining intensity categories and their respective intensity scores, using the following formula: H-score = [1 × (% of GDF-11 positive cells with intensity score 1)] + [2 × (% of GDF-11 positive cells with intensity score 2)] + [3 × (% of GDF-11 positive cells with intensity score 3)]. The GDF-11-positive cell percentage for each intensity category used in the above formula was calculated as the average of ten randomly chosen high-power fields at 400× magnification. The final H-score ranged from 0 to 300. As a consequence, while discerning between background staining and genuine positivity may not be easily facilitated under low-power magnification, finer details necessitate higher magnification. Therefore, we examine our samples at 400× magnification to reliably distinguish positive staining cells from background artifacts. Furthermore, through the utilization of H-score calculation, we maintain confidence in the capacity of our methodology to accurately quantify GDF-11 expression within our prepared tissue samples.

(g) Please, provide proper staining results with positive, negative control, faint, moderate and strong immunostainings additionally to what you have already shown.

Author Response: We have included illustrations of positive, negative control, faint, moderate, and strong immunostainings in new Figure 1 and appended explanatory sentences in the Materials and Methods section to delineate the methodologies employed for positive and negative control. Please see page 4, lines157-159 of the revised manuscript.

(2) Study cohort (p3/15 2.3. clinicopathological tumour characteristics; p4/15: 3.

Results 164-169)

(a) The study cohort is low with 68 patients.

Author Response: Thank you for the reviewer comments. It is true that the number of research cases in this study is limited. There is also limited research on the association between GDF11 and breast cancer recently. Our study is a pilot study; despite the small sample size, we believe the preliminary findings of our study still hold potential significance for advancing medical knowledge. We plan to further enlarge our sample size and investigate more details regarding the role of GDF11 in breast cancer carcinogenesis in the future. We have addressed this issue in the limitations section of the discussion. Please see page 13, lines 430-431 of the revised manuscript.

(b) The percentage of HER2-positive patients is very high with 40%. The range of HER- 2 positive cases in a cohort of newly diagnosed breast cancer patients is around 12-15%. Can you state on that?

Author Response: When we reexamined our data to define HER2-positive cases, we identified an error in categorizing cases as HER2-positive with FISH negativity. Consequently, we revised and recalculated our data. The percentage of HER2-positive patients is 25.0%. This finding aligns with a previous study (4) which indicated that breast cancer is a highly heterogeneous tumor, with HER2-positive breast cancer accounting for approximately 25-30% of incidences. We have revised Tables 1 and 2, as well as to the Abstract and Results section. Please see pages 6, 8, page 1, line 38, page 5, lines 201-202, and page 7, line 234 of the revised manuscript.

(c) How did you define luminal A and B (Ki67, ER/PR)?

Author Response: We define luminal A and B breast cancer subtypes according to the criteria established in a previous study (5): Luminal A subtype is characterized by estrogen receptor (ER) and progesterone receptor (PR) positivity (≥20%), human epidermal growth factor receptor 2 (HER2) negativity, and low Ki-67 expression (<14%). The Luminal B-like subtype, which is HER2 negative, is defined by ER positivity, HER2 negativity, and at least one of the following: high Ki-67 expression (≥14%), or negative or low PR expression (<20%). The Luminal B-like subtype with HER2 positivity is characterized by ER positivity, HER2 overexpression or amplification, any level of Ki-67 expression, and any PR expression. We have included the definitions of luminal A and B subtypes, including criteria for Ki67 expression, ER, and PR status, in the Materials and Methods section. Please see page 3, lines 126-133 of the revised manuscript.

(3) Correlation with blood parameters

(a) Correlation with blood parameters should be omitted in this study, since it does not bring additional information to the key findings and the conclusions on the association with GDF11 are very hypothetic.

Author Response: Thank you for your comments, while our study primarily focuses on the expression of GDF11 in breast cancer patients' tumor characteristics, the overarching aim of our research remains centered on investigating the impact of these fundamental tumor characteristics on breast cancer carcinogenesis, patient prognosis, and clinical progression. We believe it is imperative to further elucidate and analyze our findings regarding GDF11 expression within the context of patients' clinical profiles to ascertain any potential correlations or causal relationships. This approach aims to mitigate confounding factors such as underlying patient conditions that may influence GDF11 expression, or conversely, the influence of GDF11 on patient outcomes. We greatly appreciate your understanding of the rationale behind our methodology.

(4) Manuscript text

The text is too long and there are too many redundancies. Please, focus on the key issues and results.

Author Response: We have amended our manuscript to concentrate on the pivotal issues and findings.

References:

  1. Zhang Y, Wei Y, Liu D, Liu F, Li X, Pan L, Pang Y, Chen D. Role of growth differentiation factor 11 in development, physiology and disease. Oncotarget. 2017;8:81604-81616.
  2. Lee HH, Wang YN, Xia W, Chen CH, Rau KM, Ye L, Wei Y, Chou CK, Wang SC, Yan M, Tu CY, Hsia TC, Chiang SF, Chao KSC, Wistuba II, Hsu JL, Hortobagyi GN, Hung MC. Removal of N-Linked Glycosylation Enhances PD-L1 Detection and Predicts Anti-PD-1/PD-L1 Therapeutic Efficacy. Cancer Cell. 2019;36:168-178.e4.
  3. Chen CC, Yu TH, Wu CC, Hung WC, Lee TL, Tang WH, Tsai IT, Chung FM, Lee YJ, Hsu CC. Loss of ficolin-3 expression is associated with poor prognosis in patients with hepatocellular carcinoma. Int J Med Sci. 2023;20:1091-1096.
  4. Yang J, Ju J, Guo L, Ji B, Shi S, Yang Z, Gao S, Yuan X, Tian G, Liang Y, Yuan P. Prediction of HER2-positive breast cancer recurrence and metastasis risk from histopathological images and clinical information via multimodal deep learning. Comput Struct Biotechnol J. 2021:20:333-342.
  5. Goldhirsch A, Winer EP, Coates AS, Gelber RD, Piccart-Gebhart M, Thürlimann B, Senn HJ; Panel members. Personalizing the treatment of women with early breast cancer: highlights of the St Gallen International Expert Consensus on the Primary Therapy of Early Breast Cancer 2013. Ann Oncol. 2013;24:2206-23.
